# Assortment Optimization Under the Mallows model

**Antoine Désir**
IEOR Department
Columbia University
antoine@ieor.columbia.edu

**Vineet Goyal**
IEOR Department
Columbia University
vgoyal@ieor.columbia.edu

**Srikanth Jagabathula**
IOMS Department
NYU Stern School of Business
sjagabat@stern.nyu.edu

**Danny Segev**
Department of Statistics
University of Haifa
segevd@stat.haifa.ac.il

## Abstract

We consider the assortment optimization problem when customer preferences follow a mixture of Mallows distributions. The assortment optimization problem focuses on determining the revenue/profit maximizing subset of products from a large universe of products; it is an important decision that is commonly faced by retailers in determining what to offer their customers. There are two key challenges: (a) the Mallows distribution lacks a closed-form expression (and requires summing an exponential number of terms) to compute the choice probability and, hence, the expected revenue/profit per customer; and (b) finding the best subset may require an exhaustive search. Our key contributions are an efficiently computable closed-form expression for the choice probability under the Mallows model and a compact mixed integer linear program (MIP) formulation for the assortment problem.

## 1   Introduction

Determining the subset (or assortment) of items to offer is a key decision problem that commonly arises in several application contexts. A concrete setting is that of a retailer who carries a large universe of products $\mathcal{U}$ but can offer only a subset of the products in each store, online or offline. The objective of the retailer is typically to choose the offer set that maximizes the expected revenue/profit[1] earned from each arriving customer. Determining the best offer set requires: (a) a demand model and (b) a set optimization algorithm. The demand model specifies the expected revenue from each offer set, and the set optimization algorithm finds (an approximation of) the revenue maximizing subset.

In determining the demand, the demand model must account for *product substitution* behavior, whereby customers substitute to an available product (say, a dark blue shirt) when her most preferred product (say, a black one) is not offered. The substitution behavior makes the demand for each offered product a function of the *entire* offer set, increasing the complexity of the demand model. Nevertheless, existing work has shown that demand models that incorporate substitution effects provide significantly more accurate predictions than those that do not. The common approach to capturing substitution is through a *choice model* that specifies the demand as the probability $\mathbb{P}(a|S)$ of a random customer choosing product $a$ from offer set $S$. The most general and popularly studied class of choice models is the rank-based class [9, 24, 12], which models customer purchase decisions through distributions over preference lists or rankings. These models assume that in each choice instance, a customer samples a preference list specifying a preference ordering over a subset of the

products, and chooses the first available product on her list; the chosen product could very well be the no-purchase option.

The general rank-based model accommodates distributions with exponentially large support sizes and, therefore, can capture complex substitution patterns; however, the resulting estimation and decision problems become computationally intractable. Therefore, existing work has focused on various parametric models over rankings. By exploiting the particular parametric structures, it has designed tractable algorithms for estimation and decision-making. The most commonly studied models in this context are the Plackett-Luce (PL) [22] model and its variants, the nested logit (NL) model and the mixture of PL models. The key reason for their popularity is that the assumptions made in these models (such as the Gumbel assumption for the error terms in the PL model) are geared towards obtaining closed-form expressions for the choice probabilities $\mathbb{P}(a|S)$. On the other hand, other popular models in the machine learning literature such as the Mallows model have largely been ignored because computing choice probabilities under these models has been generally considered to be computationally challenging, requiring marginalization of a distribution with an exponentially-large support size.

In this paper, we focus on solving the assortment optimization problem under the Mallows model. The Mallows distribution was introduced in the mid-1950's [17] and is the most popular member of the so-called distance-based ranking models, which are characterized by a modal ranking $\omega$ and a concentration parameter $\theta$. The probability that a ranking $\sigma$ is sampled falls exponentially as $e^{-\theta \cdot d(\sigma, \omega)}$. Different distance functions result in different models. The Mallows model uses the Kendall-Tau distance, which measures the number of pairwise disagreements between the two rankings. Intuitively, the Mallows model assumes that consumer preferences are concentrated around a central permutation, with the likelihood of large deviations being low.

We assume that the parameters of the model are given. Existing techniques in machine learning may be applied to estimate the model parameters. In settings of our interest, data are in the form of choice observations (item $i$ chosen from offer set $S$), which are often collected as part of purchase transactions. Existing techniques focus on estimating the parameters of the Mallows model when the observations are complete rankings [8], partitioned preferences [14] (which include top-$k$/bottom-$k$ items), or a general partial-order specified in the form of a collection of pairwise preferences [15]. While the techniques based on complete rankings and partitioned preferences don't apply to this context, the techniques proposed in [15] can be applied to infer the model parameters.

**Our results.** We address the two key computational challenges that arise in solving our problem: (a) efficiently computing the choice probabilities and hence, the expected revenue/profit, for a given offer set $S$ and (b) finding the optimal offer set $S^*$. Our main contribution is to propose two alternate procedures to to efficiently compute the choice probabilities $\mathbb{P}(a|S)$ under the Mallows model. As elaborated below, even computing choice probabilities is a non-trivial computational task because it requires marginalizing the distribution by summing it over an exponential number of rankings. In fact, computing the probability of a general partial order under the Mallows model is known to be a #P hard problem [15, 3]. Despite this, we show that the Mallows distribution has rich combinatorial structure, which we exploit to derive a *closed-form expression* for the choice probabilities that takes the form of a discrete convolution. Using the fast Fourier transform, the choice probability expression can be evaluated in $O(n^2 \log n)$ time (see Theorem 3.2), where $n$ is the number of products. In Section 4, we exploit the repeated insertion method (RIM) [7] for sampling rankings according to the Mallows distribution to obtain a dynamic program (DP) for computing the choice probabilities in $O(n^3)$ time (see Theorem 4.2). The key advantage of the DP specification is that the choice probabilities are expressed as the unique solution to a system of linear equations. Based on this specification, we formulate the assortment optimization problem as a compact mixed linear integer program (MIP) with $O(n)$ binary variables and $O(n^3)$ continuous variables and constraints. The MIP provides a framework to model a large class of constraints on the assortment (often called "business constraints") that are necessarily present in practice and also extends to mixture of Mallows model. Using a simulation study, we show that the MIP provides accurate assortment decisions in a reasonable amount of time for practical problem sizes.

The exact computation approaches that we propose for computing choice probabilities are necessary building blocks for our MIP formulation. They also provide computationally efficient alternatives to computing choice probabilities via Monte-Carlo simulations using the RIM sampling method. In fact, the simulation approach will require exponentially many samples to obtain reliable estimates when

products have exponentially small choice probabilities. Such products commonly occur in practice (such as the tail products in luxury retail). They also often command high prices because of which discarding them can significantly lower the revenues.

**Literature review.** A large number of parametric models over rankings have been extensively studied in the areas of statistics, transportation, marketing, economics, and operations management (see [18] for a detailed survey of most of these models). Our work particularly has connections to the work in machine learning and operations management. The existing work in machine learning has focused on designing computationally efficient algorithms for estimating the model parameters from commonly available observations (complete rankings, top-$k$/bottom-$k$ lists, pairwise comparisons, etc.). The developed techniques mainly consist of efficient algorithms for computing the likelihood of the observed data [14, 11] and sampling techniques for sampling from the distributions conditioned on observed data [15, 20]. The Plackett-Luce (PL) model, the Mallows model, and their variants have been, by far, the most studied models in this literature. On the other hand, the work in operations management has mainly focused on designing set optimization algorithms to find the best subset efficiently. The multinomial logit (MNL) model has been the most commonly studied model in this literature. The MNL model was made popular by the work of [19] and has been shown by [25] to be equivalent to the PL model, introduced independently by Luce [16] and Plackett [22]. When given the model parameters, the assortment optimization problem has been shown to be efficiently solvable for the MNL model by [23], variants of the nested logit (NL) model by [6, 10], and the Markov chain model by [2]. The problem is known to be hard for most other choice models [4], so [13] studies the performance of the local search algorithm for some of the assortment problems that are known to be hard. As mentioned, the literature in operations management has restricted itself to only those models for which choice probabilities are known to be efficiently computed. In the context of the literature, our key contribution is to extend a popular model in the machine learning literature to choice contexts and the assortment problem.

## 2 Model and problem statement

**Notation.** We consider a universe $\mathscr{U}$ of $n$ products. In order to distinguish products from their corresponding ranks, we let $\mathscr{U} = \{a_1, \ldots, a_n\}$ denote the universe of products, under an arbitrary indexing. Preferences over this universe are captured by an anti-reflexive, anti-symmetric, and transitive relation $\succ$, which induces a total ordering (or ranking) over all products; specifically, $a \succ b$ means that $a$ is preferred to $b$. We represent preferences through rankings or permutations. A complete ranking (or simply a ranking) is a bijection $\sigma \colon \mathscr{U} \to [n]$ that maps each product $a \in \mathscr{U}$ to its rank $\sigma(a) \in [n]$, where $[j]$ denotes the set $\{1, 2, \ldots, j\}$ for any integer $j$. Lower ranks indicate higher preference so that $\sigma(a) < \sigma(b)$ if and only if $a \succ_\sigma b$, where $\succ_\sigma$ denotes the preference relation induced by the ranking $\sigma$. For simplicity of notation, we also let $\sigma_i$ denote the product ranked at position $i$. Thus, $\sigma_1 \sigma_2 \cdots \sigma_n$ is the list of the products written by increasing order of their ranks. Finally, for any two integers $i \leq j$, let $[i, j]$ denote the set $\{i, i+1, \ldots, j\}$.

**Mallows model.** The Mallows model is a member of the distance-based ranking family models [21]. This model is described by a location parameter $\omega$, which denotes the central permutation, and a scale parameter $\theta \in \mathbb{R}_+$, such that the probability of each permutation $\sigma$ is given by

$$\lambda(\sigma) = \frac{e^{-\theta \cdot d(\sigma, \omega)}}{\psi(\theta)},$$

where $\psi(\theta) = \sum_\sigma \exp(-\theta \cdot d(\sigma, \omega))$ is the normalization constant, and $d(\cdot, \cdot)$ is the Kendall-Tau metric of distance between permutations defined as $d(\sigma, \omega) = \sum_{i<j} \mathbb{1}[(\sigma(a_i) - \sigma(a_j)) \cdot (\omega(a_i) - \omega(a_j)) < 0]$. In other words, the $d(\sigma, \omega)$ counts the number of pairwise disagreements between the permutations $\sigma$ and $\omega$. It can be verified that $d(\cdot, \cdot)$ is a distance function that is right-invariant under the composition of the symmetric group, i.e., $d(\pi_1, \pi_2) = d(\pi_1 \pi, \pi_2 \pi)$ for every $\pi, \pi_1, \pi_2$, where the composition $\sigma\pi$ is defined as $\sigma\pi(a) = \sigma(\pi(a))$. This symmetry can be exploited to show that the normalization constant $\psi(\theta)$ has a closed-form expression [18] given by $\psi(\theta) = \prod_{i=1}^n \frac{1 - e^{-i \cdot \theta}}{1 - e^{-\theta}}$. Note that $\psi(\theta)$ depends only on the scale parameter $\theta$ and does not depend on the location parameter. Intuitively, the Mallows model defines a set of consumers whose preferences are "similar", in the sense of being centered around a common permutation, where the probability for

deviations thereof are decreasing exponentially. The similarity of consumer preferences is captured by the Kendall-Tau distance metric.

**Problem statement.** We first focus on efficiently computing the probability that a product $a$ will be chosen from an offer set $S \subseteq \mathscr{U}$ under the Mallows model. When offered the subset $S$, the customer is assumed to sample a preference list according to the Mallows model and then choose the most preferred product from $S$ according to the sampled list. Therefore, the probability of choosing product $a$ from the offer set $S$ is given by

$$\mathbb{P}(a|S) = \sum_\sigma \lambda(\sigma) \cdot \mathbb{1}[\sigma, a, S], \tag{1}$$

where $\mathbb{1}[\sigma, a, S]$ indicates whether $\sigma(a) < \sigma(a')$ for all $a' \in S$, $a' \neq a$. Note that the above sum runs over $n!$ preference lists, meaning that it is a priori unclear if $\mathbb{P}(a|S)$ can be computed efficiently.

Once we are able to compute the choice probabilities, we consider the assortment optimization problem. For that, we extend the product universe to include an additional product $a_q$ that represents the outside (no-purchase) option and extend the Mallows model to $n + 1$ products. Each product $a$ has an exogenously fixed price $r_a$ with the price $r_q$ of the outside option set to 0. Then, the goal is to solve the following decision problem:

$$\max_{S \subseteq \mathscr{U}} \sum_{a \in S} \mathbb{P}(a|S \cup \{r_q\}) \cdot r_a. \tag{2}$$

The above optimization problem is hard to approximate within $O(n^{1-\epsilon})$ under a general choice model [1].

# 3   Choice probabilities: closed-form expression

We now show that the choice probabilities can be computed efficiently under the Mallows model. Without loss of generality, we assume from this point on that the products are indexed such that the central permutation $\omega$ ranks product $a_i$ at position $i$, for all $i \in [n]$. The next theorem shows that, when the offer set is contiguous, the choice probabilities enjoy a rather simple form. Using these expressions as building blocks, we derive a closed-form expressions for general offer sets.

**Theorem 3.1 (Contiguous offer set)** *Suppose $S = a_{[i,j]} = \{a_i, \dots, a_j\}$ for some $1 \leq i \leq j \leq n$. Then, the probability of choosing product $a_k \in S$ under the Mallows model with location parameter $\omega$ and scale parameter $\theta$ is given by*

$$\mathbb{P}(a_k|S) = \frac{e^{-\theta \cdot (k-i)}}{1 + e^{-\theta} + \cdots + e^{-\theta \cdot (j-i)}}.$$

The proof of Theorem 3.1 is in Appendix A. The expression for the choice probability under a general offer set is more involved. For that, we need the following additional notation. For a pair of integers $1 \leq m \leq q \leq n$, define

$$\psi(q, \theta) = \prod_{s=1}^q \sum_{\ell=0}^{s-1} e^{-\theta \cdot \ell} \quad \text{and} \quad \psi(q, m, \theta) = \psi(m, \theta) \cdot \psi(q - m, \theta).$$

In addition, for a collection of $M$ discrete functions $h_m : \mathbb{Z} \to \mathbb{R}, m = 1, \dots, M$ such that $h_m(r) = 0$ for any $r < 0$, their discrete convolution is defined as

$$(h_1 \star \cdots \star h_m)(r) = \sum_{\substack{r_1, \dots, r_M: \\ \sum_m r_m = r}} h_1(r_1) \cdots h_M(r_M).$$

**Theorem 3.2 (General offer set)** *Suppose $S = a_{[i_1, j_1]} \cup \cdots \cup a_{[i_M, j_M]}$ where $i_m \leq j_m$ for $1 \leq m \leq M$ and $j_m < i_{m+1}$ for $1 \leq m \leq M - 1$. Let $G_m = a_{[j_m, i_{m+1}]}$ for $1 \leq m \leq M - 1$, $G = G_1 \cup \cdots \cup G_M$, and $C = a_{[i_1, j_M]}$. Then, the probability of choosing $a_k \in a_{[i_\ell, j_\ell]}$ can be written as*

$$\mathbb{P}(a_k|S) = e^{-\theta \cdot (k-i_1)} \cdot \frac{\prod_{m=1}^{M-1} \psi(|G_m|, \theta)}{\psi(|C|, \theta)} \cdot (f_0 \star \tilde{f}_1 \star \cdots \star \tilde{f}_\ell \star f_{\ell+1} \star \cdots \star f_M)(|G|),$$

*where:*

- $f_m(r) = e^{-\theta \cdot r \cdot (j_m - i_1 + 1 + r/2)} \cdot \frac{1}{\psi(|G_m|, r, \theta)}$, *if* $0 \le r \le |G_m|$, *for* $1 \le m \le M$.

- $\tilde{f}_m(r) = e^{\theta \cdot r} \cdot f_m(r)$, *for* $1 \le m \le M$.

- $f_0(r) = \psi(|C|, |G| - r, \theta) \cdot \frac{e^{\theta \cdot (|G| - r)^2 / 2}}{1 + e^{-\theta} + \cdots + e^{-\theta \cdot (|S| - 1 + r)}}$, *for* $0 \le r \le |G|$.

- $f_m(r) = 0$, *for* $0 \le m \le M$ *and any* $r$ *outside the ranges described above.*

**Proof.** At a high level, deriving the choice probability expression for a general offer set involves breaking down the probabilistic event of choosing $a_k \in S$ into simpler events for which we can use the expression given in Theorem 3.1, and then combining these expressions using the symmetries of the Mallows distribution.

For a given vector $R = (r_0, \ldots, r_M) \in \mathbb{R}^{M+1}$ such that $r_0 + \ldots r_M = |G|$, let $h(R)$ be the set of permutations which satisfy the following two conditions: $i)$ among all the products of $S$, $a_k$ is the most preferred, and $ii)$ for all $m \in [M]$, there are exactly $r_m$ products from $G_m$ which are preferred to $a_k$. We denote this subset of products by $\tilde{G}_m$ for all $m \in [M]$. This implies that there are $r_0$ products from $G$ which are less preferred than $a_k$. With this notation, we can write

$$\mathbb{P}(a_k | S) = \sum_{R: r_0 + \ldots r_M = |G|} \sum_{\sigma \in h(R)} \lambda(\sigma), \text{ where recall that } \lambda(\sigma) = \frac{e^{-\theta \cdot \sum_{i,j} \xi(\sigma, i, j)}}{\psi(\theta)}$$

with $\xi(\sigma, i, j) = \mathbb{1}[(\sigma(a_i) - \sigma(a_j)) \cdot (\omega(a_i) - \omega(a_j)) < 0]$. For all $\sigma$, we break down the sum in the exponential as follows: $\sum_{i,j} \xi(\sigma, i, j) = C_1(\sigma) + C_2(\sigma) + C_3(\sigma)$, where $C_1(\sigma)$ contains pairs of products $(i, j)$ such that $a_i \in \tilde{G}_m$ for some $m \in [M]$ and $a_j \in S$, $C_2(\sigma)$ contains pairs of products $(i, j)$ such that $a_i \in \tilde{G}_m$ for some $m \in [M]$ and $a_j \in G_{m'} \setminus \tilde{G}_{m'}$ for some $m \ne m'$, and $C_3(\sigma)$ contains the remaining pairs of products. For a fixed $R$, we show that $C_1(\sigma)$ and $C_2(\sigma)$ are constant for all $\sigma \in h(R)$.

*Part 1.* $C_1(\sigma)$ counts the number of disagreements (i.e., number of pairs of products that are oppositely ranked in $\sigma$ and $\omega$) between some product in $S$ and some product in $\tilde{G}_m$ for any $m \in [M]$. For all $m \in [M]$, a product in $a_i \in \tilde{G}_m$ induces a disagreement with all product $a_j \in S$ such that $j < i$. Therefore, the sum of all these disagreements is equal to,

$$C_1(\sigma) = \sum_{m=1}^{M} \sum_{\substack{a_j \in S \\ a_i \in \tilde{G}_m}} \xi(\sigma, i, j) = \sum_{m=1}^{M} r_m \sum_{j=1}^{m} |S_j|,$$

where $S_m = a_{[i_m, j_m]}$.

*Part 2.* $C_2(\sigma)$ counts the number of disagreements between some product in any $\tilde{G}_m$ and some product in any $G_{m'} \setminus \tilde{G}_{m'}$ for $m' \ne m$. The sum of all these disagreements is equal to,

$$C_2(\sigma) = \sum_{m \ne m'} \sum_{\substack{a_i \in \tilde{G}_m \\ a_j \in G_{m'} \setminus \tilde{G}_{m'}}} \xi(\sigma, i, j) = \sum_{m=2}^{M} r_m \cdot \sum_{j=1}^{m-1} (|G_j| - r_j)$$

$$= \sum_{m=2}^{M} r_m \cdot \sum_{j=1}^{m-1} |G_j| - \sum_{m=2}^{M} r_m \cdot \sum_{j=1}^{m-1} r_j$$

$$= \sum_{m=2}^{M} r_m \sum_{j=1}^{m-1} |G_j| - \frac{1}{2}(|G| - m_0)^2 + \frac{1}{2} \sum_{m=1}^{M} r_m^2.$$

Consequently, for all $\sigma \in h(R)$, we can write $d(\sigma, \omega) = C_1(R) + C_2(R) + C_3(\sigma)$ and therefore,

$$\mathbb{P}(a_k | S) = \sum_{R: r_0 + \cdots + r_M = |G|} \frac{e^{-\theta \cdot (C_1(R) + C_2(R))}}{\psi(\theta)} \cdot \sum_{\sigma \in h(R)} e^{-\theta . C_3(\sigma)}.$$

Computing the inner sum requires a similar but more involved partitioning of the permutations as well as using Theorem 3.1. The details are presented in Appendix B. In particular, we can show that for a fixed $R$, $\sum_{\sigma \in h(R)} e^{-\theta \cdot C_3(\sigma)}$ is equal to

$$\psi(|G| - m_0, \theta) \cdot \psi(|S| + m_0, \theta) \cdot \frac{e^{-\theta \cdot (k-1-\sum_{m=1}^{\ell-1} r_m)}}{1 + \cdots + e^{-\theta \cdot (|S|+m_0-1)}} \cdot \prod_{m=1}^{M} \frac{\psi(|G_m|, \theta)}{\psi(r_m, \theta) \cdot \psi(|G|_m - r_m, \theta)}.$$

Putting all the pieces together yields the desired result. ∎

Due to representing $\mathbb{P}(a_k|S)$ as a discrete convolution, we can efficiently compute this probability using fast Fourier transform in $O(n^2 \log n)$ time [5], which is a dramatic improvement over the exponential sum (1) that defines the choice probabilities. Although Theorem 3.2 allows us to compute the choice probabilities in polynomial time, it is not directly useful in solving the assortment optimization problem under the Mallows model. To this end, we present an alternative (and slightly less efficient) method for computing the choice probabilities by means of dynamic programming.

## 4   Choice probabilities: a dynamic programming approach

In what follows, we present an alternative algorithm for computing the choice probabilities. Our approach is based on an efficient procedure to sample a random permutation according to a Mallows distribution with location parameter $\omega$ and scale parameter $\theta$. The random permutation is constructed sequentially using a repeated insertion method (RIM) as follows. For $i = 1, \ldots, n$ and $s = 1, \ldots, i$, insert $a_i$ at position $s$ with probability $p_{i,s} = e^{-\theta \cdot (i-s)}/(1 + e^{-\theta} + \cdots + e^{-\theta \cdot (i-1)})$.

**Lemma 4.1 (Theorem 3 in [15])** *The random insertion method generates a random sample from a Mallows distribution with location parameter $\omega$ and scale parameter $\theta$.*

Based on the correctness of this procedure, we describe a dynamic program to compute the choice probabilities of a general offer set $S$. The key idea is to decompose these probabilities to include the position at which a product is chosen. In particular, for $i \leq k$ and $s \in [k]$, let $\pi(i, s, k)$ be the probability that product $a_i$ is chosen (i.e., first among products in $S$) at position $s$ after the $k$-th step of the RIM. In other words, $\pi(i, s, k)$ corresponds to a choice probability when restricting $\mathcal{U}$ to the first $k$ products, $a_1, \ldots, a_k$. With this notation, we have for all $i \in [n]$, $\mathbb{P}(a_i|S) = \sum_{s=1}^{n} \pi(i, s, n)$.

We compute $\pi(i, s, k)$ iteratively for $k = 1, \ldots, n$. In particular, in order to compute $\pi(i, s, k+1)$, we use the correctness of the sampling procedure. Specifically, starting from a permutation $\sigma$ that includes the products $a_1, \ldots, a_k$, the product $a_{k+1}$ is inserted at position $j$ with probability $p_{k+1,j}$, and we have two cases to consider.

**Case 1: $a_{k+1} \notin S$.**   In this case, $\pi(k+1, s, k+1) = 0$ for all $s = 1, \ldots, k+1$. Consider a product $a_i$ for $i \leq k$. In order for $a_i$ to be chosen at position $s$ after $a_{k+1}$ is inserted, one of the following events has to occur: $i)$ $a_i$ was already chosen at position $s$ before $a_{k+1}$ is inserted, and $a_{k+1}$ is inserted at a position $\ell > s$, or $ii)$ $a_i$ was chosen at position $s-1$, and $a_{k+1}$ is inserted at a position $\ell \leq s-1$. Consequently, we have for all $i \leq k$

$$\pi(i, s, k+1) = \sum_{\ell=s+1}^{k+1} p_{k+1,\ell} \cdot \pi(i, s, k) + \sum_{\ell=1}^{s-1} p_{k+1,\ell} \cdot \pi(i, s-1, k)$$
$$= (1 - \gamma_{k+1,s}) \cdot \pi(i, s, k) + \gamma_{k+1,s-1} \cdot \pi(i, s-1, k),$$

where $\gamma_{k,s} = \sum_{\ell=1}^{s} p_{k,\ell}$ for all $k, s$.

**Case 2 : $a_{k+1} \in S$.**   Consider a product $a_i$ with $i \leq k$. This product is chosen at position $s$ only if it was already chosen at position $s$ and $a_{k+1}$ is inserted at a position $\ell > s$. Therefore, for all $i \leq k$, $\pi(i, s, k+1) = (1 - \gamma_{k+1,s}) \cdot \pi(i, s, k)$. For product $a_{k+1}$, it is chosen at position $s$ only if all products $a_i$ for $i \leq k$ are at positions $\ell \geq s$ and $a_{k+1}$ is inserted at position $s$, implying that

$$\pi(k+1, s, k+1) = p_{k+1,s} \cdot \sum_{i \leq k} \sum_{\ell=s}^{n} \pi(i, \ell, k).$$

Algorithm 1 summarizes this procedure.

---
**Algorithm 1** Computing choice probabilities
---
1: Let $S$ be a general offer set. Without loss of generality, we assume that $a_1 \in S$.
2: Let $\pi(1,1,1) = 1$.
3: For $k = 1, \ldots, n-1$,
   (a) For all $i \leq k$ and $s = 1, \ldots k+1$, let

$$\pi(i, s, k+1) = (1 - \gamma_{k+1,s}) \cdot \pi(i, s, k) + \mathbb{1}[a_{k+1} \notin S] \cdot \gamma_{k+1,s-1} \cdot \pi(i, s-1, k).$$

   (b) For $s = 1, \ldots, k+1$, let

$$\pi(k+1, s, k+1) = \mathbb{1}[a_{k+1} \in S] \cdot p_{k+1,s} \cdot \sum_{i \leq k} \sum_{\ell = s}^{n} \pi(i, \ell, k).$$

4: For all $i \in [n]$, return $\mathbb{P}(a_i | S) = \sum_{s=1}^{n} \pi(i, s, n)$.
---

**Theorem 4.2** *For any offer set $S$, Algorithm 1 returns the choice probabilities under a Mallows distribution with location parameter $\omega$ and scale parameter $\theta$.*

This dynamic programming approach provides an $O(n^3)$ time algorithm for computing $\mathbb{P}(a|S)$ for all products $a \in S$ simultaneously. Moreover, as explained in the next section, these ideas lead to an algorithm to solve the assortment optimization problem.

## 5 Assortment optimization: integer programming formulation

In the assortment optimization problem, each product $a$ has an exogenously fixed price $r_a$. Moreover, there is an additional product $a_q$ that represents the outside option (no-purchase), with price $r_q = 0$ that is always included. The goal is to determine the subset of products that maximizes the expected revenue, i.e., solve (2). Building on Algorithm 1 and introducing a binary variable for each product, we formulate 2 as an MIP with $O(n^3)$ variables and constraints, of which only $n$ variables are binary.

We assume for simplicity that the first product of $S$ (say $a_1$) is known. Since this product is generally not known a-priori, in order to obtain an optimal solution to problem (2), we need to guess the first offered product and solve the above integer program for each of the $O(n)$ guesses. We note that the MIP formulation is quite powerful and can handle a large class of constraints on the assortment (such as cardinality and capacity constraints) and also extends to the case of the mixture of Mallows model.

**Theorem 5.1** *Conditional on $a_1 \in S$, the optimal solution to 2 is given by $S^* = \{i \in [n] \colon x_i^* = 1\}$, where $\boldsymbol{x}^* \in \{0,1\}^n$ is the optimal solution to the following MIP:*

$$\max_{\boldsymbol{x}, \boldsymbol{\pi}, \boldsymbol{y}, \boldsymbol{z}} \quad \sum_{i,s} r_i \cdot \pi(i, s, n)$$

$$\begin{aligned}
\text{s.t.} \quad & \pi(1,1,1) = 1, \pi(1,s,1) = 0, && \forall s = 2, \ldots, n \\
& \pi(i, s, k+1) = (1 - w_{k+1,s}) \cdot \pi(i, s, k) + y_{i,s,k+1}, && \forall i, s, \forall k \geq 2 \\[1em]
& \pi(k+1, s, k+1) = z_{s,k+1}, && \forall s, \forall k \geq 2 \\
& y_{i,s,k} \leq \gamma_{k+1,s-1} \cdot \pi(i, s-1, k-1), && \forall i, s, \forall k \geq 2 \\
& 0 \leq y_{i,s,k} \leq \gamma_{k+1,s-1} \cdot (1 - x_k), && \forall i, s, \forall k \geq 2 \\
& z_{s,k} \leq p_{k+1,s} \cdot \sum_{\ell=s}^{n} \sum_{i=1}^{k-1} \pi(i, \ell, k-1), && \forall s, \forall k \geq 2 \\
& 0 \leq z_{s,k} \leq p_{k+1,s} \cdot x_k, && \forall s, \forall k \geq 2 \\
& x_1 = 1, x_q = 1, x_k \in \{0,1\}
\end{aligned}$$

We present the proof of correctness for this formulation in Appendix C.

# 6 Numerical experiments

In this section, we examine how the MIP performs in terms of the running time. We considered the following simulation setup. Product prices are sampled independently and uniformly at random from the interval $[0, 1]$. The modal ranking is fixed to the identity ranking with the outside option ranked at the top. The outside option being ranked at the top is characteristic of applications in which the retailer captures a small fraction of the market and the outside option represents the (much larger) rest of the market. Because the outside option is always offered, we need to solve only a single instance of the MIP (described in Theorem 5.1). Note that in the more general setting, the number of MIPs that must be solved is equal the minimum of the rank of the outside option and the rank of the highest revenue item[2]. Because the MIPs are independent of each other, they can be solved in parallel. We solved the MIPs using the Gurobi Optimizer version 6.0.0 on a computer with processor 2.4GHz Intel Core i5, RAM of 8GB, and operating system Mac OSX El Capitan.

**Strengthening of the MIP formulation.** We use structural properties of the optimal solution to tighten some of the upper bounds involving the binary variables in the MIP formulation. In particular, for all $i, s,$ and $m$, we replace the constraint $y_{i,s,m} \leq \gamma_{m+1,s-1} \cdot (1 - x_m)$ with the constraint $y_{i,s,m} \leq \gamma_{m+1,s-1} \cdot u_{i,s,m} \cdot (1 - x_m)$, where $u_{i,s,m}$ is the probability that product $a_i$ is selected at position $(s - 1)$ after the $m^{th}$ step of the RIM when the offer set is $S = \{a_{i^*}, a_q\}$, i.e. when only the highest priced product is offered. Since we know that the highest price product is always offered in the optimal assortment, this is a valid upper bound to $\pi(i, s - 1, m - 1)$ and, therefore, a valid strengthening of the constraint. Similarly, for all $s$ and $m$, we replace the constraint, $z_{s,m} \leq \alpha_{m+1,s} \cdot x_m$ with the constraint $z_{s,m} \leq \alpha_{m+1,s} \cdot v_{s,m} \cdot x_m$, where $v_{s,m}$ is equal to the probability that product that product $i$ is selected at position $\ell = s, \ldots, n$ when the offer set is $S = \{a_q\}$ if $a_i \succ_w a_{i^*}$, and $S = \{a_q, a_{i^*}\}$ otherwise. Again using the fact that the highest price product is always offered in the optimal assortment, we can show that this is a valid upper bound.

**Results and discussion.** Table 1 shows the running time of the strengthened MIP formulation for different values of $e^{-\theta}$ and $n$. For each pair of parameters, we generated 50 different instances. We

| $n$ | Average running time (s) | | Max running time (s) | |
|---|---|---|---|---|
| | $e^{-\theta} = 0.8$ | $e^{-\theta} = 0.9$ | $e^{-\theta} = 0.8$ | $e^{-\theta} = 0.9$ |
| 10 | 4.60 | 4.72 | 5.64 | 5.80 |
| 15 | 19.04 | 21.30 | 27.08 | 28.79 |
| 20 | 48.08 | 105.30 | 58.09 | 189.93 |
| 25 | 143.21 | 769.78 | 183.78 | 1,817.98 |

Table 1: Running time of the strengthened MIP for various values of $e^{-\theta}$ and $n$.

note that the strengthening improves the running time considerably. Under the initial formulation, the MIP did not terminate after several hours for $n = 25$ whereas it was able to terminate in a few minutes with the additional strengthening. Our MIP obtains the optimal solution in a reasonable amount of time for the considered parameter values. Outside of this range, i.e. when $e^{-\theta}$ is too small or when $n$ is too large, there are potential numerical instabilities. The strengthening we propose is one way to improve the running time of the MIP but other numerical optimization techniques may be applied to improve the running time even further. Finally, we emphasize that the MIP formulation is necessary because of its flexibility to handle versatile business constraints (such as cardinality or capacity constraints) that naturally arise in practice.

**Extensions and future work.** Although the entire development was focused on a single Mallows model, our results extend to a finite mixture of Mallows model. Specifically, for a Mallows model with $T$ mixture components, we can compute the choice probability by setting $\pi(i, s, n) = \sum_{t=1}^{T} \alpha_t \cdot \pi_t(i, s, n)$, where $\pi(i, s, n)$ is the probability term defined in Section 4, $\pi_t(\cdot, \cdot, \cdot)$ is the probability for mixture component $t$, and $\alpha_t > 0$ are the mixture weights. We then have $\mathbb{P}(a|S) = \sum_{s=1}^{n} \pi(i, s, n)$ for the mixture model. Correspondingly, the MIP in Section 5 also naturally extends. The natural next step is to develop special purpose algorithm to solve the MIP that exploit the structure of the Mallows distributions allowing to scale to large values of $n$.

## Footnotes

[1] As elaborated below, conversion-rate maximization can be obtained as a special case of revenue/profit maximization by setting the revenue/profit of all the products to be equal.

[2]It can be shown that the highest revenue item is always part of the optimal subset.

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
