[Supplementary Material]

# A Proof of Theorem 3.1

In order to prove Theorem 3.1, it suffices to show that for any $k \in [i, j-1]$, $\mathbb{P}(a_k|S) = e^{-\theta} \cdot \mathbb{P}(a_{k+1}|S)$. For a fixed $k$, let $A = \{\sigma : a_k \succ_\sigma a_j, \forall a_j \in S \backslash \{a_k\}\}$ and $B = \{\sigma : a_{k+1} \succ_\sigma a_j, \forall a_j \in S \backslash \{a_{k+1}\}\}$. Let $f : A \to B$ be the function which switches the position of $a_k$ and $a_{k+1}$. Note that it is a bijection from $A$ to $B$. Consequently, we can write

$$\mathbb{P}(a_k|S) = \sum_{\sigma \in A} \frac{e^{-\theta \cdot d(\sigma,w)}}{\psi(\theta)} \quad \text{and} \quad \mathbb{P}(a_{k+1}|S) = \sum_{\sigma \in A} \frac{e^{-\theta \cdot d(f(\sigma),w)}}{\psi(\theta)}.$$

We next show that for all $\sigma \in A$, $d(f(\sigma), w) = d(\sigma, w) + 1$, which in turn implies the desired result. Because $S$ is a contiguous set, $a_k$ and $a_{k+1}$ are consecutive items in $w$. This implies that in any fixed $\sigma$, any disagreement between $a_k$ and some $a_j$ with $a_j \neq a_{k+1}$ will induce a disagreement between $a_{k+1}$ and $a_j$ in $f(\sigma)$. Similarly, any disagreement between $a_{k+1}$ and some $a_j$ with $a_j \neq a_k$ in $\sigma$ will induce a disagreement between $a_k$ and $a_j$ in $f(\sigma)$. Consequently, the only additional disagreement in $f(\sigma)$ comes from the disagreement between $a_k$ and $a_{k+1}$ after being switched. This implies that for all $\sigma \in A$, $d(f(\sigma), w) = d(\sigma, w) + 1$ and concludes the proof.

# B Proof of Theorem 3.2 (continued)

In this section, we prove that for a fixed $R$, $\sum_{\sigma \in h(R)} e^{-\theta \cdot C_3(\sigma)}$ is equal to

$$\psi(|G| - m_0, \theta) \cdot \psi(|S| + m_0, \theta) \cdot \frac{e^{-\theta \cdot (k - 1 - \sum_{m=1}^{\ell-1} r_m)}}{1 + \cdots + e^{-\theta \cdot (|S| + m_0 - 1)}} \cdot \prod_{m=1}^{M} \frac{\psi(|G_m|, \theta)}{\psi(r_m, \theta) \cdot \psi(|G|_m - r_m, \theta)}.$$

We use a similar approach than in the first part of the proof. Let $\Gamma$ be the set of $(\tilde{G}_1, \ldots, \tilde{G}_M) \subseteq (G_1, \ldots, G_M)$ such that $|\tilde{G}_m| = r_m$ for all $m \in [M]$. For all $\gamma = (\tilde{G}_1, \ldots, \tilde{G}_M) \in \Gamma$, let $t(\gamma)$ be the set of permutations $\sigma$ which satisfy the following two conditions:

- $\sigma \in h(R)$.
- for all $m \in [M]$, the subset of products from $G_m$ which is preferred to $a_k$ is exactly $\tilde{G}_m$.

With this notation, we can write

$$\sum_{\sigma \in h(R)} e^{-\theta \cdot C_3(\sigma)} = \sum_{\gamma \in \Gamma} \sum_{\sigma \in t(\gamma)} e^{-\theta \cdot (D_1(\sigma) + D_2(\sigma) + \sum_{m \in [M]} D_3(\sigma,m))},$$

where,

- $D_1(\sigma)$ is the sum of disagreements $\xi(\sigma, i, j)$ over pairs of products $(i, j)$ such that either $i = k$ and $a_k \succ_\sigma a_j$ or $a_k \succ_\sigma a_i$ and $a_k \succ_\sigma a_j$.
- $D_2(\sigma)$ is the sum of disagreements $\xi(\sigma, i, j)$ over pairs of products $(i, j)$ such that $a_i \succ_\sigma a_k$ and $a_j \succ_\sigma a_k$.
- for all $m \in [M]$, $D_3(\sigma, m)$ is the sum of disagreements $\xi(\sigma, i, j)$ over pairs of products $(i, j)$ such that $a_i \in \tilde{G}_m$ and $a_j \in G_m \backslash \tilde{G}_m$.

Using the definition of $D_1(\sigma)$ and $D_2(\sigma)$ together with Theorem 3.1, we have that $\sum_{\sigma \in t(\gamma)} e^{-\theta \cdot (D_1(\sigma) + D_2(\sigma) + \sum_{m \in [M]} D_3(\sigma,m))}$ is equal to

$$\psi(|G| - m_0, \theta) \cdot \psi(|S| + m_0, \theta) \cdot \frac{e^{-\theta \cdot (k - 1 - \sum_{m=1}^{\ell-1} r_m)}}{1 + \cdots + e^{-\theta \cdot (|S| + m_0 - 1)}} \cdot \sum_{\sigma \in t(\gamma)} e^{-\theta \cdot \sum_{m \in [M]} D_3(\sigma,m)}.$$

To complete the proof, it remains to compute $\sum_{\gamma \in \Gamma} \sum_{\sigma \in t(\gamma)} e^{-\theta \cdot \sum_{m \in [M]} D_3(\sigma,m)}$. Using the definition of the normalization constant, we have for all $m \in [M]$,

$$\psi(|G_m|, \theta) = \psi(r_m, \theta) \cdot \psi(|G_m| - r_m, \theta) \cdot \sum_{\gamma \in \Gamma} \sum_{\sigma \in t(\gamma)} e^{-\theta \cdot D_3(\sigma,m)},$$

which implies that

$$\sum_{\gamma \in \Gamma} \sum_{\sigma \in t(\gamma)} e^{-\theta \cdot \sum_{m \in [M]} D_3(\sigma,m)} = \prod_{m=1}^{M} \frac{\psi(|G_m|, \theta)}{\psi(r_m, \theta) \cdot \psi(|G_m| - r_m, \theta)},$$

and concludes the proof.

## C   Proof of Theorem 5.1

Let $x = (x_1, \ldots, x_n)$ be a feasible binary vector to the IP and let $S = \{a_i : x_i = 1\}$. Note that there is a one to one correspondence between feasible vector $x$ to the IP and feasible assortment $S$ such that $a_1 \in S$ and $a_q \in S$. In particular, $x_i = 1$ if $i \in S$ and $x_i = 0$ otherwise. Consequently, we can rewrite the IP as

$$
\max_{\substack{S \subseteq \mathscr{U} \\ a_q \in S}} \max \quad \sum_{i,s} r_i \cdot \pi(i, s, n)
$$

$$
\begin{aligned}
\text{s.t.} \quad & \pi(i, s, k+1) = (1 - w_{k+1,s}) \cdot \pi(i, s, k) + y_{i,s,k+1}, && \forall i, s, \forall k \geq 2 \\
& \pi(k+1, s, k+1) = z_{s,k+1}, && \forall s, \forall k \geq 2 \\
& 0 \leq y_{i,s,k} \leq \mathbf{1}[a_{k+1} \notin S] \cdot \gamma_{k+1,s-1} \cdot \pi(i, s-1, k-1), && \forall i, s, \forall k \geq 2 \\
& 0 \leq z_{s,k} \leq \mathbf{1}[a_{k+1} \in S] \cdot p_{k+1,s} \cdot \sum_{\ell=s}^{n} \sum_{i=1}^{k-1} \pi(i, \ell, k-1), && \forall s, \forall k \geq 2 \\
& \pi(1, 1, 1) = 1
\end{aligned}
$$

Note that it is always optimal to set $y_{i,s,k}$ and $z_{s,k}$ at their upper bound because all the coefficients in the objective function are non-negative. The correctness of Algorithm 1 then shows that the IP is an equivalent formulation of (2).