[Reviews · NeurIPS 2016]

Reviewer 1

Summary

The paper has three main contributions: a) the choice probability is computed for arbitrary set under Mallows model in a closed form. The key observation here is that the choice probability for a contiguous set is rather simple to compute based the definition of the Mallows model (Theorem 3.1). Then using this observation choice probability is computed by decomposing the choice set into the union of contiguous sets. b) the choice probabilities are computed for Mallows model by using dynamic programming based on the fact that the Mallows model is a repeated insertion model. c) based on the dynamic programming formulation, the assortment problem is solved via a mixed integer linear program. The constraint of the linear program is computed based on the dynamic programming solution. Synthetic experiments are carried out to asses the wall-clock running time of the linear programming solver with respect to the number of the products and the parameter of the underlying Mallows model.

Qualitative Assessment

1) Even if Theorem 3.1 is basically coming from the definition of the model, at least a short explanation or proof might be added. 2) Regarding the proof of Theorem 3.2 a) the term ``products'' was slightly misleading. Why is not ``element'' used? S is in fact a set. b) the decomposition step is nice and well-explained c) Regarding the comment of Fast Fourier transformation: I am not sure that the choice probability can be computed based in O (n^2 \log n) time. Should not it be O( n^2 \log^2 n )? (because there are three products are nested when computing P( a | S ) ) 3) There are two different methods are given to compute the same quantity, namely, the choice probabilities for Mallows model. What is the relation of them? I mean the dynamic programing computation is, in fact, doable because of the L-decomposability of Mallows model (see Doignon et. al. 2004). But what is the key property of Mallows model which allows us to have write the choice probabilities in a closed-form? 4) The linear programming task is not clear for me. What is the relation of the LP described in the appendix and the one described in Theorem 5.1. In particular, what I would like to understand how one can get rid of the max over the power set of S. Could you please elaborate more on this issue? Except Section 5, the paper is easy to follow. p.2 line 65 to to -> to

Confidence in this Review

3-Expert (read the paper in detail, know the area, quite certain of my opinion)


Reviewer 2

Summary

This paper focuses on solving the assortment optimization problem in the context of the Mallows model. The authors addressed two challenges: (i) computing the choice probabilities and expected return for a offer set efficiently (ii) finding the optimal offer set.

Qualitative Assessment

The methods and mathematical formalization seem sound, but I am not quite satisfied with the experiments. Table 1 (MIP runtime) is not very convincing. The parameters grid is not consistent in terms of scale. Would be good if the authors could provide a runtime (line) chart showing the trend instead of a table. Another weak point of the experiment is: the proposed methods are introduced as a tool for retail applications. However, the experiments only showed a max. of n=20, which is a small number to be considered as a practical method for real world retail applications (which usually have thousands of products).

Confidence in this Review

2-Confident (read it all; understood it all reasonably well)


Reviewer 3

Summary

The author propose methods to efficiently compute the choice probabilities under the Mallows model. They further formulate a compact mixed integer linear program for the assortment problem. They provide numerical experiments that demonstrate the performance of the proposed solution.

Qualitative Assessment

Potential impact / Technical quality: I think this is an interesting paper, however it could be significantly improved by including additional experiments on real world data that demonstrate its effectiveness for real world scenarios. Since the authors explicitly states in their rebuttal that their main contribution is theoretical, this work can be seen as a first step in a promising research direction.

Confidence in this Review

2-Confident (read it all; understood it all reasonably well)


Reviewer 4

Summary

This paper gives a closed form expression for computing choice probabilities under the Mallows model. Further, it gives a mixed integer linear programming formulation for the assortment optimization problem for the Mallows model.

Qualitative Assessment

Contribution of the paper is limited to giving a efficiently computable closed form expression for choice probabilities under the Mallows model, and a sampling based method to compute the same.

Confidence in this Review

2-Confident (read it all; understood it all reasonably well)


Reviewer 5

Summary

This paper studies an assortment optimization problem under the Mallow distribution model. It first offers a closed form expression for computing the choice probability of an item among a certain assortment that relies on discrete convolution. The authors argue that this result implies that the probability can be obtained efficiently using a Fourier transform method. On the other hand, they reject this closed form expression in their attempt to solve the assortment optimization where they rely on a recursive computation method to formulate a mixed integer linear programming problem formulation. With this formulation, they show that they have the capacity of solving problems with 20 items.

Qualitative Assessment

In my opinion, the authors do not provide convincing enough arguments that computing choice probabilities is as hard as they let it appear. The main reference that is stated relates to the difficulty of « computing the probability of a general parital order » which is not what is achieved by the methods that are proposed here. Instead, based on the repeated insertion procedure that is described in this paper (and obtained from Doignon et al. 2004) it appears that this probability can be efficiently computed using Monte-Carlo simulations (in fact, such simulations could even be used to construct a sample average approximation of problem (2)). I agree with the authors that the two methods that they propose are more efficient but have difficulty estimating the size of the contribution based on the text. Perhaps some information about sample average statistical convergence or some numerical evaluation of the methods would provide such an argument. Regarding the second contribution, while I found the derivation of the MILP cleverly done. It fells short of convincing the reader that the model as real practical appeal. Indeed, in practice one would suspect that assortment planning have to handle numbers of items that are 10 to 100 times bigger than what the authors were able to achieve. One would also suspect that solutions to the assortment problem must be made almost instantaneously as users are visiting a web site and information are gathered about them. Overall, I felt that there is a disconnect between section 3 and the rest of the paper that distracts the message that is made. Moreover, I believe that the MILP formulation has value but further efforts should be deployed in order to improve numerical performances if the goal is to have a practical impact with it. I would also encourage the author to identify a journal or conference that is more strongly oriented towards operations research or operations management. Finally, here are some minor things that could be corrected : - Page 2, line 65 : « to to efficiently » should be « to efficiently » - Page 4, line 161 : « choosing a_k \in a_{i_l,j_l} » you should also define « l » here - Page 6, line 211 : « at position s » should sat « and at position s » - Page 7, line 228 : I don’t see how you can assume that a_1 \in S here given that a_1 is the preferred item in U. Please explain how the algorithm needs to change when it is not the case. Do we simply discard all items from U that are preferred to all items in S. - Page 8, line 280 : By « add cutting planes », I assume you mean « add valid inequalities » so that the LP relaxation is tighter. Or do you propose employing a decomposition scheme : e.g. master problem only holds the x_i’s while the slave problem takes care of the z’s and y’s

Confidence in this Review

2-Confident (read it all; understood it all reasonably well)


Reviewer 6

Summary

The paper describes a choice problem and provides a solution under the Mallows model with Kendall distance.

Qualitative Assessment

I have no many concerns with the paper. I consider interesting to know when it is possible to extend these results to the generalised Mallows models. Another point that requires justification is the knowledge of the Mallows distribution that generated the data. One point that should be cleared is the comment the authors do in relation to "...The above optimization problem is hard to approximate within O(n1 ✏) under general choice model [1]..." How does this relate with their approach?

Confidence in this Review

3-Expert (read the paper in detail, know the area, quite certain of my opinion)